# Thiourea Derivatives, Simple in Structure but Efficient Enzyme Inhibitors and Mercury Sensors

**DOI:** 10.3390/molecules26154506

**Published:** 2021-07-27

**Authors:** Faizan Ur Rahman, Maryam Bibi, Ezzat Khan, Abdul Bari Shah, Mian Muhammad, Muhammad Nawaz Tahir, Adnan Shahzad, Farhat Ullah, Muhammad Zahoor, Salman Alamery, Gaber El-Saber Batiha

**Affiliations:** 1Department of Chemistry, University of Malakand, Lower Dir 18800, Pakistan; faizan.uom98@gmail.com (F.U.R.); mianchem@uom.edu.pk (M.M.); 2Department of Chemistry, College of Science, University of Bahrain, Sakhir 32038, Bahrain; maryamdua55@gmail.com; 3Division of Applied Life Science (BK21 Plus), IALS, Gyeongsang National University, Jinju 52828, Korea; abs.uom28@gmail.com; 4Department of Physics, University of Sargodha, Sargodha 40100, Pakistan; dmntahir_uos@yahoo.com; 5Institute of Chemical Sciences, University of Swat, Swat 19150, Pakistan; adnanshahzad09@gmail.com; 6Department of Pharmacy, University of Malakand, Chakdara, Lower Dir 18800, Pakistan; farhataziz80@hotmail.com; 7Department of Biochemistry, University of Malakand, Lower Dir 18800, Pakistan; mohammadzahoorus@yahoo.com; 8Department of Biochemistry, College of Science, King Saud University, P.O. Box 22452, Riyadh 11451, Saudi Arabia; Salamery@ksu.edu.sa; 9Department of Pharmacology and Therapeutics, Faculty of Veterinary Medicine, Damanhour University, Damanhour 22511, Egypt; gaberbatiha@gmail.com

**Keywords:** thiourea derivatives, enzyme inhibition, docking studies, mercury sensing, X-ray structure

## Abstract

In this study six unsymmetrical thiourea derivatives, 1-isobutyl-3-cyclohexylthiourea (**1**), 1-*tert-*butyl-3-cyclohexylthiourea (**2**), 1-(3-chlorophenyl)-3-cyclohexylthiourea (**3**), 1-(1,1-dibutyl)-3-phenylthiourea (**4**), 1-(2-chlorophenyl)-3-phenylthiourea (**5**) and 1-(4-chlorophenyl)-3-phenylthiourea (**6**) were obtained in the laboratory under aerobic conditions. Compounds **3** and **4** are crystalline and their structure was determined for their single crystal. Compounds **3** is monoclinic system with space group P2_1_/n while compound **4** is trigonal, space group *R^3^:H*. Compounds (**1**–**6**) were tested for their anti-cholinesterase activity against acetylcholinesterase and butyrylcholinesterase (hereafter abbreviated as, AChE and BChE, respectively). Potentials (all compounds) as sensing probes for determination of deadly toxic metal (mercury) using spectrofluorimetric technique were also investigated. Compound **3** exhibited better enzyme inhibition IC_50_ values of 50, and 60 µg/mL against AChE and BChE with docking score of −10.01, and −8.04 kJ/mol, respectively. The compound also showed moderate sensitivity during fluorescence studies.

## 1. Introduction

Thiourea derivatives play a pivotal role due to numerous applications in the field of nanoparticles, starting precursors in a large number of chemical reactions and therefore, remained focus of several review articles [1,2,3,4,5]. Efficient and easy accessibility of thiourea derivatives is an open challenge [6]. Several researchers have made thiourea derivatives under different conditions and many of them are now commercially available [7,8]. Thiourea derivatives such as cyclohexyl thiourea or phenylthiourea are attractive model compounds for the studies in solid-state chemistry due to their tendency for the formation of intra- and inter molecular hydrogen bonding wherein the N-H proton-donor groups and predominantly sulfur atoms are engaged. The presence of C=S groups in the same molecule make them more attractive for further chemistry [9]. These derivatives have an active role in pharmaceuticals as potential therapeutic agents against HIV, HCV [10,11,12,13], anticancer, anticonvulsant [14,15], antihyperlipidemic, antiallergic, antiparasitic, antiproliferative, antioxidant and antidiabetic and anticholinestrase [16,17]. Cholinesterases are various isolated enzymes which are classified as acetyl and butyryl associated with hydrolases enzymes [18,19]. Acetylcholinesterase (AChE) provides a signaling action following degradation of acetylcholine (ACh) [20,21], whereas butyrylcholinesterase (BChE) plays a pivotal role in the metabolism of different molecules. Hence, AChE inhibitors are considered to be one of the most efficient approaches for the treatment of Alzheimer’s disease (AD) by enhancing cholinergic objectives in AD patients. In a normal healthy brain, ACh metabolic degradation is also caused by BChE, hence BChE enhances progression of AD. Therefore, synchronous inhibition achieved by AChE and BChE may provide additional benefits for AD treatment [22]. Several thiourea derivatives such as isobutylphenylthiourea, and tert-butylphenylthiourea and their coordination complexes have recently been reported by our research group as enzyme inhibitors [23]. Several studies have been conducted with respect to thiourea derivatives which highlight the role of this versatile class of compounds in environmental applications particularly heavy metal sensing [24,25,26,27,28,29,30]. Heavy metals are expected to be one of the reasons of the AD and their timely control in the body of an individual is utmost necessary [31,32]. 

Fluorescence is one of the most important physical phenomena in which a particular chemical compound (thiourea for instance) emits electromagnetic radiations of longer wavelength after being excited by radiations with shorter wavelength. Fluorescence based detection, is a homogeneous system that allows direct measurement of binding in solution and has played a significant role at the forefront of the bio-analytical area, because of ultrahigh sensitivity and selectivity. One of the most important applications of fluorescence technology is the decreased size of a sample down to the single-molecule detection level, and this fact provides an opportunity for miniaturization and high throughput screening. In addition, fluorescence has enabled the elucidation of many biological processes, determination of structure and trace amounts of biological macromolecules. Fluorescence sensors employ a fluorescent signal for the detection of analytes. In order to detect different organics at an earlier stage of a disease development, a low detection limit of the organics is desirable [33]. Thiourea derivatives possess the ability to detect both anions and cations and are therefore attracting the attention to be versatile detectors in coming decades [5]. Thiourea derivatives are used as fluorescent detectors for heavy metal ions including Hg^2+^ and various anions in aqueous media [4,34,35,36].

The AD is greatly influenced by heavy metal ions concentration and can be controlled either by inhibition of specific enzymes or quenching of responsible metal ions. In the current paper, we present the dual function of thiourea derivatives for the same purpose to control AD in an efficient way. Literature review reveals that solid state structure of these compounds is not yet reported. We hereby report solid state structure of compounds **3** and **4**, enzyme inhibition and fluorescence potentials of thiourea derivatives which are already in field (Figure 1). It is notable that most of the compounds are potent inhibitors with respect to the standard, galantamine. Molecular docking study is also carried out for compound **3** and **4** in order to investigate their inhibitory action. The synthesized compounds were also investigated as a sensing probe for determination of mercury with the intent to use these biologically active molecules in removal technology in order to eliminate complications of bioacceptance and accessibility. The fluorescent properties of compounds are very interesting and the compound **2** was employed in the determination of mercury, as sensing probe. These studies invite further research in the field of single molecule detectors for determination of heavy metal ions of biological concerns in controlling a variety of diseases. 

## 2. Results and Discussion

Compounds **1**–**6** are structurally very simple; they are accessible in the market and can also be obtained in laboratory under ambient conditions [37]. Compounds structurally analogous to **1**–**6** are good corrosion inhibitors in acidic medium [38] and show excited results in affording metal sulfides for useful applications [39]. Compound **5** has been used as starting precursors for the preparation of benzothiazole derivatives under catalyst free conditions [40]. There are several other reports wherein simple thiourea molecules after certain modifications have shown efficiency as bioactive compounds [41]. Compound **6** has already been used for its inhibitory activity against melanin B16 cells and mushroom tyrosinase and its synthesis has been reviewed [6,42]. The IC_50_ value as melanin B16 inhibitor was promising, 3.4 μM, while it exhibited moderate potency as mushroom tyrosinase inhibitor. The enzyme inhibition (AChE and BChE) and metal sensing capability of compounds **1**–**6** (Figure 1) have not been reported so far. Since they are already in the field of bioorganic chemistry therefore these studies will help exploring their multidimensionality applications within a single system or organism whatever the case may be. 

### 2.1. Structural Description of Compound ***3***

Compound **3** was crystalline in nature and good quality crystals were obtained at room temperature, which allowed for collection of X-ray diffraction data. The structure of compound **3** is depicted in Figure 2, along with a dimer unit held together with the help of H-bonding. The refinements and crystal structure solution parameters are summarized in Table 1, Bond angles and lengths of structural importance are summarized in Table 2, and the experimental data are compared with theoretical. Both data sets were in close agreement with each other. The surrounding of C7 is trigonal planar with expected bond lengths and angles and the data also revealed the sp^2^ hybridized environment of the carbon. The C=S bond length was 1.695 Å, which fall within the expected limit for the said derivative [43]. The C7-N1 and C7-N2 distances were 1.351 and 1.326 Å, where the effect of substituent on the N was very clear, the lone pair of N1 was delocalized towards Ph ring while that of N2 was attracted towards C7 imparting partial double bond characters to N2-C7 bond (Table 2). This difference in delocalization of electron pair on both the N atoms provided insights to identify suitable H-bonding sites. The 3-ClPh-N1 was comparatively electron rich and is suitable donor, the hydrogen bonding was thus established making a dimer as given in Figure 2. Short ranged interactions were not observed in molecules of compound **3**. See Table 3 for data pertaining to hydrogen bond parameters. 

### 2.2. Structural Description of Compound ***4***

Molecular structure of thiourea derivative **4** is given in Figure 3 (left), structure determination and refinements were carried out as summarized in Table 1, and selected bond lengths and angles are summarized in Table 2. The bond angles and bond lengths around the central carbon, C7 were very similar to compound **3**. The sum of all bond angles is 360° indicating that C7 was sp^2^ hybridized with characteristic trigonal planar geometry. The C7=S1 bond length was 1.686 Å, which is slightly shorter than the same bond in compound **3**. The C7-N1 and C7-N2 distances were 1.352 and 1.344 Å were slightly different from each other revealing the effect of substituent on N atom. The difference was small as compared to compound **3**, indicating that lone pair of electrons at both N atoms were almost equally involved in delocalization through the NCN fragment. The NH and CH_2_hydrogen atoms in molecules played an important role in stabilizing the supramolecular structure in a 3D manner. The H8B and NH1 were attached to S1 of one molecule with H–S separation distance of 2.748 and 2.704 Å, respectively and the H8A to S atom of the other molecule (separation distance 2.960 Å). The difference in separation distance was obvious and it can be clearly observed that the former interaction combined molecule in hexameric ring form and the latter one helped to keep chains together in a 3D fashion. Non-covalent interactions between molecules of a hexameric ring made an $R_{2}^{1}\left(7\right)$ type loop in a cyclic arrangement and further extended in a 3D manner as discussed earlier. Different views of supramolecular structure of compound **4** are shown in Figure 4, and the hydrogen bonding data are presented in Table 3.

Optimized structure of compound **3** and **4**, are given in Figure 5, a comparison of experimental and theoretical bond parameters showed a close relationship, with negligible variation. The data suggested that gas phase and solid-state structure did not vary from each other. 

### 2.3. Enzyme Inhibition Studies 

The effectiveness of the synthesized compounds **1**–**6** as inhibitors of the selected cholinesterase was tested as discussed in experimental section, vide supra. The observed activities are summarized in Table 4, The IC_50_ value of most of the compounds was high > 100 µg/mL, except two compounds. Compound **3** was the most potent with excellent activity 50 and 60 µg/mL against AChE and BChE, respectively. Compound **4** also exhibited very close IC_50_ values 58 and 63 µg/mL, respectively. The IC_50_ value observed for the standard was 15 µg/mL for inhibition of both the enzymes in a respective manner. Further derivatization of compound **3**, can be useful in future studies to achieve the target properties. The efficiency of this compound can be linked to the intermolecular interactions as discussed above. Lack of secondary interactions in other molecules like **4** is the probable reason of poor biological efficiency.

#### Molecular Docking Studies of Compounds **3** and **4**

The acetylcholinesterase, responsible for hydrolysis of neurotransmitter acetylcholine (ACh) in neuromuscular junctions, consisted of two active sites, namely peripheral site and catalytic site. To corelate the obtained inhibition results in experiments, the molecular docking investigations were carried out against the selective and potent inhibitors with AChE and BChE, by following the literature procedure. The in vitro study showed that the compound **3** is the excellent and selective inhibitor of AChE and BChE supporting the experimental data of enzyme inhibition given in Table 4. To study the binding extent and binding poses of selective thiourea derivatives **3** and **4,** the active pockets were selected for binding interactions. The binding interaction of compound **3** and **4** with AChE and BChE are given in Figure 6.

The complete analysis of co-crystalized thiourea derivative **3** and AChE inhibitor showed that tyr-124, Ser-203 and Tyr-337 are the most potent residues showing binding interactions with inhibitor compound **3** with in the active pocket of protein. Our synthesized compound **3** forms strong hydrogen bonding with Tyr-124 with calculated length 2.4 Å and with binding score of −10.01 kJ/mol. In addition to strong hydrogen bonding, some weak polar interactions and hydrophobic interactions were also observed with Tyr-337, Tyr-341 and Ser-203 given in Figure 6a. The compound **4** forms hydrogen bond with Trp-532 with a separation distance of 2.7 Å with binding score of −8.031 kJ/mol and hydrophobic interactions with His-405 and Pro-537 given Figure 6b. The binding energy shows that compound **3** exhibited strong binding interactions as compared with compound **4**, thus supporting the experimental inhibition data of structurally analogous compounds [44].

During the docking studies of compound **3** and **4** with BChE, it was observed that **3** established interactions with Asn-83, Asn-85 and His-77, while compound **4** was efficient in establishing interactions with Tyr-128 and Trp-82. These interactions are responsible to make a compound more potent against BChE. The co-crystalized compound **3** formed strong hydrogen bonding with Asn-83 and His-126 with a distance of 2.2 and 3.0 Å, respectively and the binding score of −8.04 kJ/mol. In addition to this, compound **3** also established weak hydrophobic interactions with same residues of BChE given in Figure 6c, while compound **4** formed weak hydrogen bonding with Tyr-456 (2.4 Å) and weak interaction with Lys-427 residues (3.2 Å) with binding score of −6.98 kJ/mol [45]. The comparison of binding energy shows that compound **3** was also a potent inhibitor of BChE as compared to compound **4**, thus our calculations support the experimental data, Figure 6d.

### 2.4. Fluorescence and UV-Visible Characteristics of the Compounds ***1***–***3***, ***5*** and ***6***

The UV-visible absorption spectra of compounds were measured for a 2 ppm solution in the range 200–800 nm. The absorption maxima of all the tested compounds appeared between 200–400 nm as shown in Appendix A. The methanol solution of compounds exhibited λ_max_ 296, 294, 298, 294 and 295 nm, respectively. The fluorescence characteristics of the synthesized compounds (**1**–**3**, **5**, **6**) were investigated against the solvent blank by scanning the λ_ex_ and λ_em_ in the range of 250–750 nm. The fluorescence spectra of compounds are shown in Figure 7 (for two selected compounds) and respective data are tabulated in Table 5. It was observed that compound **3**, **5** and **6** showed excellent fluorescence intensity, compound **3** at low sensitivity level while compounds **1** and **2** show excellent instrumental sensitivity. The large stokes shift (>300 nm) for compounds **3** and **5** could possibly be attributed to the intramolecular charge transfer characteristic (ICT) due to the electron-accepting nature of the moiety because of its well-defined dipole moment. Compounds which undergo ICT, exhibited different properties both in their ground and excited states, which was caused by an electronic rearrangement after photo excitation. This temporary dipole alternation led to larger changes in dipole moments and to the relaxation of the structure. The energy difference between the Frank–Condon state and the ICT state was the predominant reason for the large Stokes shift, which is ideal in fluorescence methods [46,47]. The fluorescence behavior of these compounds shows that all the compounds at concentration as low as 10 µg mL^−1^ could be used for fabrication of fluorescence-based sensing tools for detection of environmental pollutants especially pesticides, toxic heavy metals and drugs. The analyte molecules could interact effectively with the compounds causing fluorescence enhancement or fluorescence quenching. The interaction could be easily employed for development of sensitive analytical methods for determination of the subject analytes which is the future perspective of this work.

### 2.5. Application of Compounds ***1***–***3***, ***5***, ***6*** as Sensing Probe for Determination of Mercury

In order to know the interaction between Hg(II) and the subject compounds, variable concentration of Hg(II) solutions was added to known concentration of each compound and FI at the respective excitation and emission of each compound was measured at low and high sensitivity mode. The results are given in Table 6 and for two selected compounds **3** and **5** are shown in Figure 8. As evident from the data, FI of the mixtures increased linearly with increase in the concentration of Hg(II) up to the maximum FI level. The reason behind this interaction of the compounds with Hg(II) was based on the soft and hard acid base concept [36]. Mercury (Hg(II)) is a soft center with high polarizability and sulfur in the thiourea molecule is soft center too. The only compound that does not show this sort of enhancement interaction with Hg(II) is **6**. Though **6** also bore a sulfur atom the probable reason of negligible interactions could be due to inappropriate cavity size and orientation for binding of Hg(II) ions. It happened due to diminishing of photoinduced electron transfer phenomenon between receptor and fluorophore [36]. This fluorescence enhancement could be successfully applied for trace level determination of Hg(II) in complex environmental samples including water, soil, vegetables and fruits. The same could also be applied for remediation of Hg(II) ion in real biological samples in order to develop efficient reagent/medicine to combat AD. 

## 3. Materials and Methods

### 3.1. General Consideration

All reactions were carried out under aerobic atmosphere and no special precautions were taken to exclude air or moisture during experiments. All chemicals were purchased from Sigma Aldrich, TCI or Across Organics and were used without further purification. Thiourea derivatives described in this study are also commercially available in the marked, we reproduced them in the laboratory, they were obtained as crystalline material. The formation of compounds was confirmed with the help of available spectroscopic techniques, such as ATR-FTIR spectrometer and ^1^H- and ^13^C NMR spectra (in order to assist readers these data is provided in Appendix A). Single crystal X-ray analysis was carried out at room temperature by using a Bruker Kappa APEXII CCD diffractometer. Absorption of solution for determination of enzyme inhibition was measured by Thermoelectron corporation, USA and florescence studies were carried out using fluorescence spectrophotometer model RF-5301 Schimadzu, Japan, for their potential use in sensing environmental pollutants. Thiourea derivatives were prepared by reactions as previously reported [44,48]. 

### 3.2. Anticholinesterase Evaluation

Anticholinesterase activity of compounds **1**–**6** was evaluated by Ellman’s assay using two enzymes, AChE and BChE and following our previously reported protocol without further modifications [49,50,51]. The enzyme AChE is derived from Electric eel and BChE from equine serum [52]. Buffer of phosphate, 0.1 M at pH 8.0 was prepared to whom varied concentration (31.625–1000 μg/mL) of the respective compound was added. The solution of AChE was diluted to 0.03 U/mL from 518 U/mg, similarly BChE from 7–16 U/mg to 0.01 U/mL at the given pH condition. Distilled water was used as solvent, for preparation of 0.5 mM acetylthiocholine iodide (ATchI), 0.2273 mM 5,5′-dithiobis-2-nitrobenzoic acid (DTNB) and 0.5 mM butyrylthiocholine iodide (BTchI). All solutions were stored at 8 °C in a refrigerator prior to performing the experiment.

Enzyme solution (5 μL), corresponding compound (205 μL), and DTNB reagent (5 μL) were mixed in a cuvette. The solution was incubated for 15 min at 30 °C followed by addition of 5 μL of the substrate solution. Absorbance of the incubated sample was measured at 412 nm by a double-beam spectrophotometer (Thermo Electron Corporation, USA) for 4 min along with the reaction time at 30 °C. The well-established Galanthamine was used as positive control and all experiments were performed in triplicate. The percent enzyme activity and enzyme inhibition by control and test samples were calculated from the rate of absorption with respect to change in time (V = ΔAbs/Δt) using the same procedure as reported previously [49,50,51].

### 3.3. Molecular Docking Study

Molecular docking was carried out to determine the binding interaction mode of the synthesized compounds for the competitive and noncompetitive inhibition with the mark enzymes, AChE and BChE [53,54]. The crystal structure of the said enzymes was obtained from protein databank. Proteins were theoretically prepared by using auto dock mgl tools [55]. and was optimized by removing all water molecules, heteroatoms and co-factors, for simplicity in calculations. The structures of the synthesized compound **3** and **4** were obtain from CIFs using Avogadro and respective CCDC software. Protein-ligands images were developed by using pymol [56,57].

### 3.4. Fluorescence Measurements

Spectroflourophotometer (RF-5301 PC, Shimadzu, Japan) having 150 W Xenon lamp as excitation source with 1.0 cm quartz cell was used throughout the experimental work for fluorescence measurements. The emission and excitation slit width was 4 nm for fluorometric operation. The thiourea derivatives (**1**–**3**, **5**, **6**) of variable concentration were prepared in acetone and incubated for 60 min at room temperature. All the tested thiourea derivatives, were found to show excellent fluorescence behavior at low as well as at high sensitivity levels. These compounds were employed to fabricate a sensing probe for determination of mercury in water samples.

## 4. Conclusions

Thiourea derivatives **1**–**6** were reproduced in the laboratory and structure of two compounds **3** and **4** was determined with the help of X-ray diffraction for single crystal. All compounds were tested for their enzyme inhibitory efficiency where compounds **3** was found more efficient. The activity of this compound (IC_50_ 50 and 60 µg/mL against AChE and BChE, respectively) is very close to the efficiency of the standard drug and is worth studying in future investigations. The experimental data particularly inhibitory efficiency of compound **3**, were validated through its docking studies and both data sets were found in close agreement with each other. Further investigations to know their sensing capacity against heavy metals ions, such as Hg(II) were carried out and the results are promising, compound **3** came out to be among other compounds with better sensitivity level. This study reveals that thiourea derivatives possess manifold useful characteristics and may be able to control AD by inhibiting AChE, BChE or controlling the concentration of heavy metal ion. These compounds are better sensors to determine specific metals in real biological samples.

## Figures and Tables

**Figure 1 molecules-26-04506-f001:**
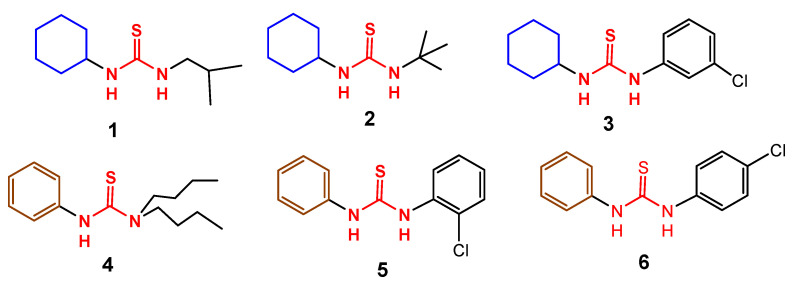
Structure of unsymmetrical thiourea derivatives, 1-cyclohexyl-3-(*iso*-butyl)thiourea (**1**), 1-cyclohexyl-3-(*tert*-butyl)thiourea (**2**), 1-cyclohexyl-3-(3-chlorophenyl)thiourea (**3**), 1-phenyl-3-(1,1-dibutyl)thiourea (**4**), 1-phenyl-3-(2-chlorophenyl)thiourea (**5**) and 1-phenyl-3-(4-chlorophenyl)thiourea (**6**) used in this study.

**Figure 2 molecules-26-04506-f002:**
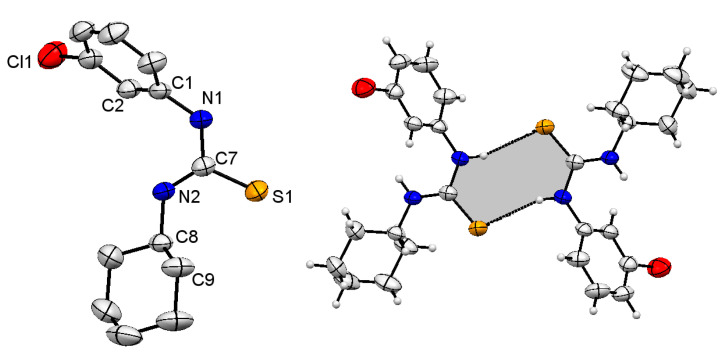
X-ray structure of Compound **3** (**left**), ellipsoids are drawn at 50% probability level, H atoms are omitted, selected carbon atoms and all non-carbon atoms are numbered. Dimer of compound **3** (**right**) stabilized through N1H–S1 interactions. Other secondary interactions in molecules were not detected. For data related to structural determination and refinements and selected structural parameters (bond lengths and angles) see Table 1 and Table 2, respectively.

**Figure 3 molecules-26-04506-f003:**
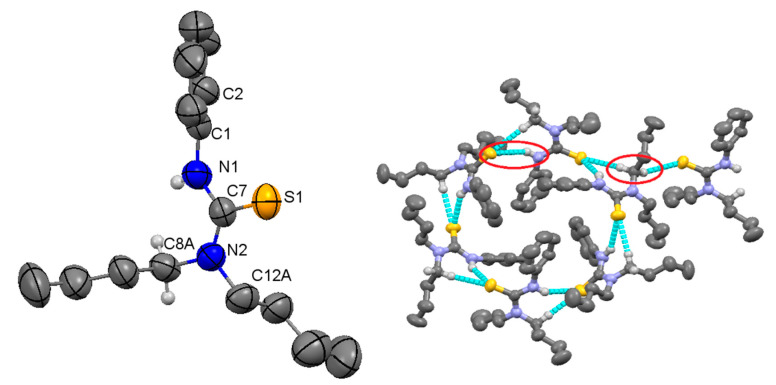
Single crystal structure (**left** hand) of compound **4**, ellipsoids are drawn at 50% probability level, H-atoms except NH and CH_2_ are omitted and selected atoms have been numbered. A cyclic hexamer (**right** hand) is shown, molecules are linked with the help of S–H interactions making a macrocyclic ring containing six thiourea molecules (See Table 2 for selected structural parameters).

**Figure 4 molecules-26-04506-f004:**
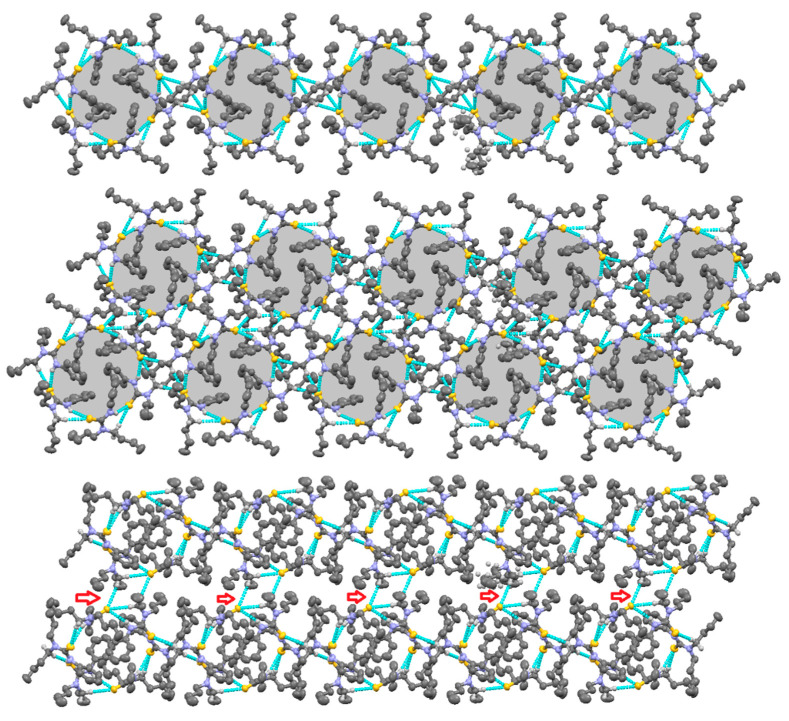
Different view of supramolecular structure of compound **4**, single layer of mutually connected hexameric units (**top**), mutually connected layers (**middle** and **bottom**), the molecular architecture extends in a 3D manner.

**Figure 5 molecules-26-04506-f005:**
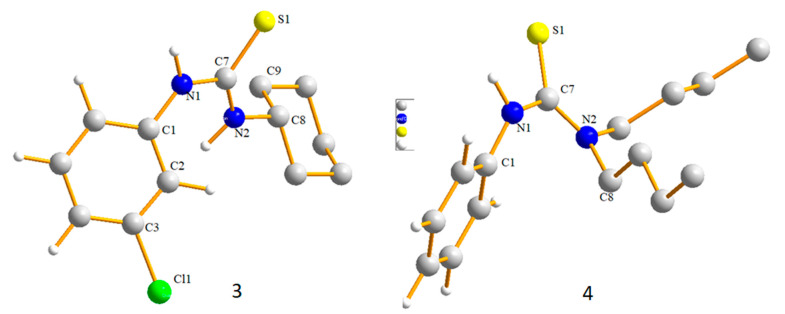
Optimized structures of compounds **3** and **4**, selected hydrogen atoms are omitted.

**Figure 6 molecules-26-04506-f006:**
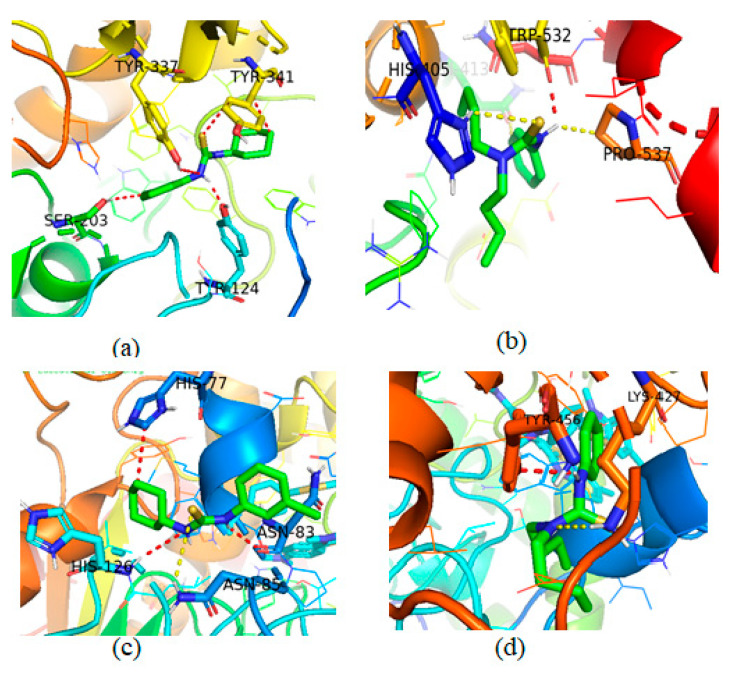
**3**D binding interaction modes of compounds **3** and **4** as inhibitor of AChE and BChE. (**a**) Interaction posing of compound **3** with AChE; (**b**) interaction of compound **4** with AChE; (**c**) interaction of compound **3** with BChE and (**d**) interaction of compound **4** with BChE.

**Figure 7 molecules-26-04506-f007:**
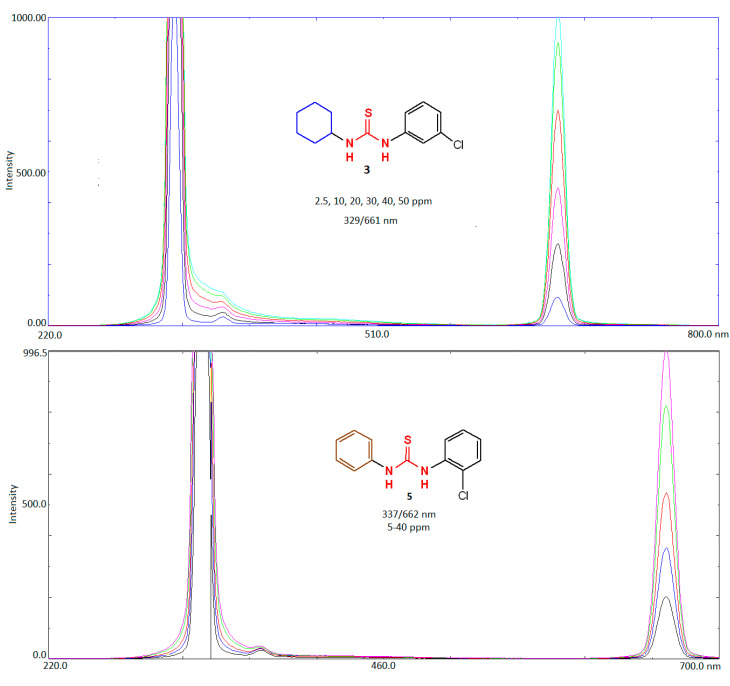
Fluorescence spectra of **3** and **5** as two representative examples.

**Figure 8 molecules-26-04506-f008:**
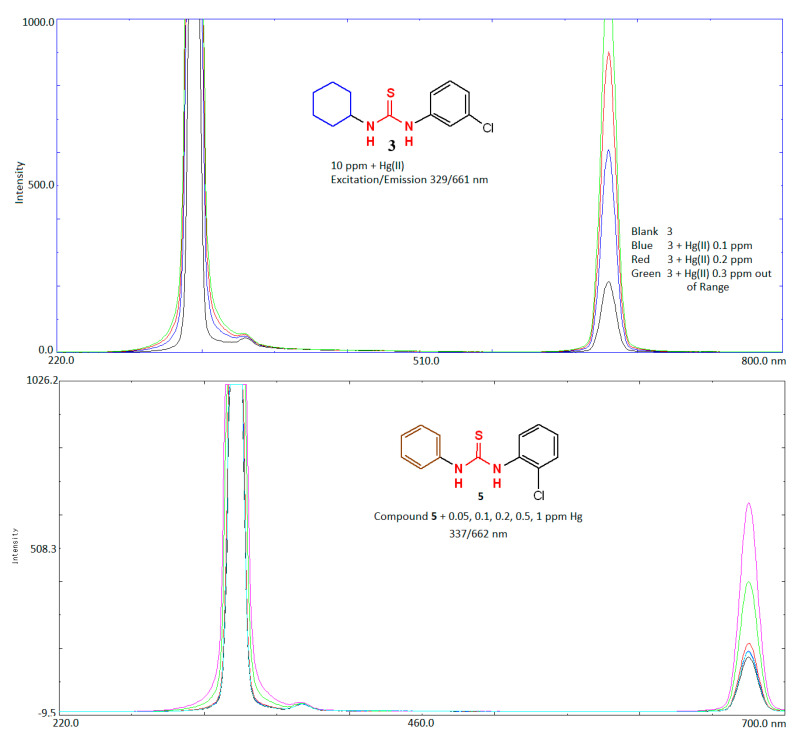
Representative fluorescence spectra of **3** and **5** as a function of concentration of Hg(II) ion.

**Table 1 molecules-26-04506-t001:** Structure solution and refinements parameters of compound **3** and **4**.

	Compound 3	Compound 4
CCDC No *	2050647	2050648
Chemical formula	C_13_H_17_ClN_2_S	C_15_H_24_N_2_S
*M* _r,_	268.79	264.42
Temperature (K)	296	296
Crystal system, space group	Monoclinic, *P*2_1_/*n*	Trigonal, *R*^3^:*H*
*a*, *b*, *c* (Å)	5.5080 (4), 22.1068 (17), 11.4404 (8)	25.550 (2), 25.550 (2), 13.1763(11),
β (°)	99.967 (4)	90.00
*V* (Å^3^)	1372.01 (17)	7448.8(13)
*Z*	4	18
Radiation type	Mo *K*α	Mo *K*α
µ (mm^−1^)	0.41	0.18
Crystal size (mm)	0.40 × 0.32 × 0.26	0.44 × 0.38 × 0.36
Diffractometer	Bruker Kappa APEXII CCD	
Absorption correction	Multi-scan	Multi-scan
T_min_, T_max_	0.815, 0.925	0.875, 0.955
No. of measured, independent and observed [*I* > 2σ(*I*)] reflections	9399, 3283, 2340	12267, 3614, 1852
*R* _int,_	0.048	0.074
(sin θ/λ) _max_ (Å^−1^)	0.660	0.641
*R*[*F*^2^ > 2σ(*F*^2^)], *wR*(*F*^2^), *S*	0.048, 0.135, 1.02	0.056, 0.168, 1.02
No. of reflections	3283	3614
No. of parameters/restrains	154/0	188/214
H-atom treatment	H-atom parameters constrained	
Δρ_max_, Δρ_min_ (e Å^−3^)	0.34, −0.32	0.18, −0.16

* Crystallographic information files pertaining to compounds **3** and **4** can be obtained free of charge at www.ccdc.cam.ac.uk/conts/retrieving.html (or from the Cambridge Crystallographic Data Centre, 12 Union Road, Cambridge CB2 1EZ, UK; Fax: +44-1223-336-033; e-mail: deposit@ccdc.cam.ac.uk).

**Table 2 molecules-26-04506-t002:** Experimental and theoretical (B3LYP/3-21G level) selected structural parameters, bond lengths and angles of compounds **3** and **4**.

Compound	Atoms	Exp.	Cal.	Atoms	Exp.	Cal.
**3**	Bond length			Bond angle		
	S1-C7	1.695(19)	1.730	C7-N1-C1	127.52(17)	128.41
	N1-C7	1.351(2)	1.383	C7-N2-C8	124.61(16)	124.81
	N1-C1	1.423(2)	1.434	N2-C7-N1	117.68(17)	117.29
	N2-C7	1.326(2)	1.342	N2-C7-S1	123.18(15)	123.53
	N2-C8	1.464(2)	1.483	N1-C7-S1	119.14(15)	118.73
**4**	S1-C7	1.686(2)	1.731	C7-N1-C1	127.84(19)	128.23
	Ni-C7	1.352(3)	1.390	N2-C7-N1	115.86(18)	116.10
	N1-C1	1.426(3)	1.421	N2-C7-S1	123.03(18)	124.34
	N2-C7	1.344(3)	1.351	C7-N2-C8B	123.6(17)	123.59

**Table 3 molecules-26-04506-t003:** Hydrogen-bond geometries (Å) of compounds **3** and **4**.

Compound	*D*—H···*A*	*D*—H	H···*A*	*D*···*A*	*D*—H···*A*	Symmetry Codes
**3**	N1—H1···S1	0.86	2.53	3.343 (18)	157.4	−x, −y, −z + 2
	C2—H2···Cl1	0.93	2.96	3.878 (2)	167.6	−x, −y, −z + 1
	C9—H9B···S1	0.97	2.91	3.483 (3)	118.9	-
**4**	N1—H1···S1	0.86	2.70	3.484 (2)	151.5	x − y + 2/3, x + 1/3, −z + 4/3
	C8A—H8A ··S1	0.97	2.96	3.840 (12)	151.5	−x + y − 1/3, −x + 1/3, z + 1/3
	C8A—H8B ··S1	0.97	2.75	3.711 (11)	172.2	x − y + 2/3, x + 1/3, −z + 4/3
	C8B—H8C ··S1	0.97	2.95	3.84 (4)	152.3	−x + y − 1/3, −x + 1/3, z + 1/3
	C8B—H8D ··S1	0.97	2.62	3.53 (4)	155.3	x − y + 2/3, x + 1/3, −z + 4/3
	C9B—H9C ··N1	0.97	2.65	3.179 (14)	114.8	-
	C12B—H12D ··S1	0.97	2.56	3.00 (10)	107.6	-

**Table 4 molecules-26-04506-t004:** Efficiency of compounds **1**–**6** as inhibitors against AChE and BChE.

No	Conc. (µg/mL)	%AChE Inhibition	IC_50_ (µg/mL)	%BChE Inhibition	IC_50_ (µg/mL)
1	1000	69.47 ± 0.22	89	74.4 ± 0.68	84
	500	63.94 ± 0.45		66.2 ± 0.73	
	250	57.61 ± 1.70		61.0 ± 0.33	
	125	53.64 ± 0.16		56.4 ± 0.63	
	62.5	46.52 ± 0.38		46.9 ± 0.42	
2	1000	63.34 ± 0.98	300	65.17 ± 0.72	185
	500	56.32 ± 1.06		57.85 ± 0.97	
	250	48.05 ± 0.75		51.37 ± 1.65	
	125	44.70 ± 1.25		46.73 ± 0.78	
	62.5	38.74 ± 0.68		41.34 ± 1.01	
3	1000	73.39. ± 0.60	50	76.7 ± 0.66	60
	500	67.39 ±0.49		71.3 ± 1.11	
	250	61.36 ± 0.49		65.5 ± 1.04	
	125	57.34 ± 0.55		57.2 ± 0.57	
	62.5	51.90 ± 1.16		49.9 ± 0.65	
4	1000	93.58 ± 1.12	58	98.00 ± 0.00	63
	500	85.40 ± 0.20		94.40 ± 0.50	
	250	77.85 ± 2.26		89.80 ± 1.50	
	125	71.80 ± 1.50		77.40 ± 1.70	
	62.5	47.90 ± 0.47		53.50 ± 0.90	
	31.5	39.01 ± 0.88		47.30 ± 0.70	
5	1000	66.79 ± 0.63	345	71.62 ± 0.74	310
	500	59.67 ± 0.61		63.86 ± 0.60	
	250	41.69 ± 0.77		44.48 ± 0.64	
	125	35.54 ± 0.50		37.54 ± 0.50	
	62.5	29.00 ± 0.30		31.74 ± 0.61	
6	1000	69.58 ± 1.12	285	71.33 ± 0.49	265
	500	61.65 ± 1.34		63.03 ± 0.23	
	250	47.90 ± 0.96		49.00 ± 0.58	
	125	39.03 ± 0.48		42.67 ± 0.89	
	62.5	31.90 ± 0.48		33.00 ± 1.15	
Galantamine	1000	83.19 ± 0.73	15	89.64 ± 0.62	15
	500	77.54 ± 0.91		81.83 ± 0.37	
	250	71.93 ± 0.14		74.29 ± 0.73	
	125	65.72 ± 0.49		67.92 ± 0.98	
	62.5	61.67 ± 0.94		64.93 ± 0.87	

**Table 5 molecules-26-04506-t005:** Fluorescence characteristics of compounds **1**–**3**, **5** and **6**.

Compound	Concentration (µg mL^−1^)	Scan Range (nm)	λ_ex_ (nm)	λ_em_ (nm)	Mode of Sensitivity	Fluorescence Intensity
**1**	10	350–700	312	613	High	276.483
**2**		220–750	304.0	328.0	High	16.189
**3**		220–650	329	661	Low	212.958
**5**		220–600	337	662	Low	180.234
**6**		220–600	311	624	Low	275.143

**Table 6 molecules-26-04506-t006:** Hg(II) sensing studies of compounds **1**–**3**, **5** and **6**, FI in the Table stands for fluorescence intensity.

Compound	Concentration (ppm)	λ_ex/_λ_em_(nm) Fluorescence Intensity	Concentration of Hg(II) ppm	Fluorescence Intensity
3	10	329/661 FI = 212.958 Curve:e	0.1	608.132
			0.2	901.543
			0.3	1015.790
1	10	312/613 FI = 276.483 Curve:e	0.1	460.580
			0.2	646.740
			0.3	876.296
			0.5	1015.790
2	0.01	264/284 FI = 16.189 Curve:e	0.01	60.193
			0.02	81.624
			0.03	117.136
			0.04	139.836
5	5.0	337/662 FI = 180 Curve:e	0.05	197
			0.1	220
			0.2	400
			0.5	550
6	10	311/624 FI = 275 Curve:e	0.05	No interaction
			0.1	
			0.2	
			1.0	

## Data Availability

The authors confirm that the data supporting the findings of this study are available within the article and its Appendix A.

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
