# Peer review of "Thiourea Derivatives, Simple in Structure but Efficient Enzyme Inhibitors and Mercury Sensors"

_molecules, 2021, doi:10.3390/molecules26154506_

Round 1
Reviewer 1 Report
Proposed manuscript has serious flaws. Synthesized compounds were already obtained in past by other researchers. Please see Reaxys or other source of information for reference. Structures were not cited and charactarized as novel. Further review comments are not necessary at this point.
Author Response
Proposed manuscript has serious flaws. Synthesized compounds were already obtained in past by other researchers. Please see Reaxys or other source of information for reference. Structures were not cited and charactarized as novel. Further review comments are not necessary at this point.
- We respect the views of the reviewer and according to literature only compound 6 is reported, which is properly cited (Ref. 6) as for as its AChE, BChE and metal sensing is concerned, we did not find any report in literature.
- Although the compounds are not really novel but this work is focused on metal sensing and inhibition of enzymes responsible for AD. On the other hand heavy metals have been reported to be the probable reason of AD. These compounds can sense the metal and can be helpful in arresting the responsible metal ion too. It is hoped that the strategy can work in future research to think on multidimensional control of an issue. The IC50 values of compound 3 in inhibiting enzymes of this study is very close to standard values that clearly indicate the applicability of thiourea derivatives to control dementia and Alzheimer disease which still needs confirmation in in vivo along with toxicological consideration, for which our research is in progress in animal models. These compounds have potential applications and are worth reporting.
Reviewer 2 Report
Rehman and co-workers prepared an interesting work but i have couple general remarks.
The synthesis of compounds and target compounds itself are very simple. There is no innovation here.
Fluorescent properties of obtained compounds are interesting but why the Stokes shift are so big? Could you explain that?
There is lack of absorbtion spectra. Emission spectra are hard to understend.
What literature says about emission of such compounds? Reference 34 is not usefull.
Author Response
Rehman and co-workers prepared an interesting work but i have couple general remarks.
The synthesis of compounds and target compounds itself are very simple. There is no innovation here.
- Novelty of a research in terms of synthesis and simplicity in structure is one aspect of the research.
- Our compounds are no doubt simple in structure but effective enzyme inhibitors (AChE and BChE), also sensing ability of compounds is promising and they are worth reporting.
- This strategy could be helpful in future researches to control an ailment in a multidimensional approach. The thiourea derivatives can sense heavy metal ions at their very low concentration level, and can arrest them to inhibit their damaging effect.
- We agree with the reviewer however, the biological and metal sensing activities about these compounds has not been reported before. The IC50 values of one of our compounds is promising and it pushes us to extend the work towards control of dementia and Alzheimer disease through thiourea derivatives. In future work we are plaining for in vivo studies along with toxicological considerations in animal models.
Fluorescent properties of obtained compounds are interesting but why the Stokes shift is so big? Could you explain that?
- The large stokes shift (>300 nm) for the representative compounds 3 and 5 can be attributed to the intramolecular charge transfer characteristic (ICT) arising due to the electron-accepting nature of the moiety because of a well-defined dipole moment. Compounds with ICT show different properties in the ground and excited states, which is caused by an electronic rearrangement after photoexcitation. This momentary alternation leads to large changes in dipole moments and to the relaxation of the structure. The energy difference between the Frank–Condon state and the ICT state is the reason for the large Stokes shift, which is ideal in most fluorescence methods. The text and two relevant references have been added to the main manuscript.
There is lack of absorbtion spectra. Emission spectra are hard to understand.
We really do not understand how to reply to this comment. Therefore, we did nothing in response to this comment, however, we would be glad to respond accordingly, if the reviewer gives us clear directions what to do?
What literature says about emission of such compounds? Reference 34 is not useful.
Relevant literature has been included in introduction sections and Yes reference 34 was irrelevant and has been replaced.
Round 2
Reviewer 1 Report
Please find comments in attachment.

Author Response
Dear Reviewer, As corresponding author I really apologize for the blunder in our writing. Since we are working with Coordination Chemistry and these compounds were supposed to be used in complexation with metal ions. Sometimes we blindly believe in our students which is not good for a professional. At the first use of these compounds, we came across their attractive properties and decided to report this dual behavior of simple thiourea derivatives. As suggested, relevant references have been added and the experimental details like synthesis and spectroscopic studies have been moved to supplementary file. The main focus of the paper remained enzyme inhibition, docking studies and Hg(II) sensing. Crystal structures of compounds 3 and 4 are not yet reported therefore, x-ray studies are also part of the paper.
I on behalf of coauthors, am personally indebted for your suggestions which are very valuable as a professional.
Reviewer 2 Report
Further, there is no information on the fluorescence of similar compounds from the literature.
There are no absorption spectra.
Why are there 2 peaks in the fluorescence spectrum? Why does one of them correspond to λex (nm)?
What is: Mode of Sensitivity, Fluorescence Intensity?
The spectra are clipped.
The paper talks about the detection of ions through fluorescence and the results are very laconic. There is not enough information about the fluorescence itself.
Author Response
Pointwise reply to the reviewer comments is given as below;
1. Further, there is no information on the fluorescence of similar compounds from the literature.
Some references wherein fluorescence of thiourea derivatives have been disclosed are mentioned in the introduction section. These articles include one of our recent review articles on thiourea derivatives for metal ion sensing.
2. There are no absorption spectra?
The absorption spectra of compounds for 2 ppm methanolic solutions are recorded and all details have been incorporated. The Figure has been included in supplimentary file as Figure S1.
3. Why are there 2 peaks in the fluorescence spectrum? Why does one of them correspond to λex (nm)?
Fluorescence is the property of some atoms and molecules to absorb light at a particular wavelength λex (nm) followed by subsequent emission of light with longer wavelength, λem (nm) after a brief interval, termed the fluorescence lifetime. Absorption of light occurs very quickly (approximately in a femtosecond, the time necessary for the photon to travel a single wavelength) in discrete amounts called quanta and corresponds to excitation of the fluorophore from the ground state to an excited state. Technically, the excitation band for any compound corresponds to absorption band.
4. What is: Mode of Sensitivity, Fluorescence Intensity?
The mode of sensitivity was high for compounds with lesser fluorescence intensity while it was kept low for highly fluorescent compounds as given qualitatively in table 5 against each compound.
5. Is the spectra are clipped.
It is evident from the appearance of the spectra that they are not clipped.
6. The paper talks about the detection of ions through fluorescence and the results are very laconic. There is not enough information about the fluorescence itself.
While basic information in research papers are not provided as such types of papers are of interest to research communities who are expert in the field and are aware about basic information. As for the results are concerned enough details are there in the text, figures and tables. Our recent review deals with sensing applications of thiourea derivatives, which is given in the list of references and interested readers can read it for more details. Here in this paper the provision of basic information will unnecessarily prolong the manuscript.
Thank you for your time and efforts.
Round 3
Reviewer 1 Report
Manuscript has been improved in line with suggestions and will be attractive to Molecules readers.
Reviewer 2 Report
Both absorbtion and emission spectra are presented not in accordance with the state of art. The range is shown wrong! The excitation signal should not be visible on the emission spectrum! I understand that the presented results may be interesting, but when we show sensing applications, let's get it right.
I still don't understand why the Stokes shifts are so large. The publications cited show something else. The general explanation is understandable, but why for some derivatives the emission is about 400 nm (literature and compound 2), and for others it is over 600 nm?